

# Latest observations on the low energy excess in CRESST-III

G. Angloher[1], S. Banik[2,3], G. Benato[4], A. Bento[1,5], A. Bertolini[1], R. Breier[6],
C. Bucci[4], L. Canonica[1], A. D'Addabbo[4], S. Di Lorenzo[4], L. Einfalt[2,3], A. Erb[7,8],
F. v. Feilitzsch[7], N. Ferreiro Iachellini[1], S. Fichtinger[2], D. Fuchs[1*], A. Fuss[2,3],
A. Garai[1], V. M. Ghete[2], S. Gerster[9], P. Gorla[4], P. V. Guillaumon[4], S. Gupta[2], D. Hauff[1],
M. Ješkovský[6], J. Jochum[9], M. Kaznacheeva[7†], A. Kinast[7‡], H. Kluck[2], H. Kraus[10],
A. Langenkämper[1,7], M. Mancuso[1], L. Marini[4,11], L. Meyer[9], V. Mokina[2], A. Nilima[1∘],
M. Olmi[4], T. Ortmann[7], C. Pagliarone[4,12], L. Pattavina[4,7], F. Petricca[1], W. Potzel[7],
P. Povinec[6], F. Pröbst[1], F. Pucci[1], F. Reindl[2,3], J. Rothe[7], K. Schäffner[1], J. Schieck[2,3],
D. Schmiedmayer[2,3], S. Schönert[7], C. Schwertner[2,3], M. Stahlberg[1], L. Stodolsky[1],
C. Strandhagen[9], R. Strauss[7], I. Usherov[7], F. Wagner[2], M. Willers[7] and V. Zema[1]

⋆ fuchsdom@mppmu.mpg.de ,    † margarita.kaznacheeva@tum.de ,
‡ angelina.kinast@tum.de ,    ∘ anilima@mpp.mpg.de

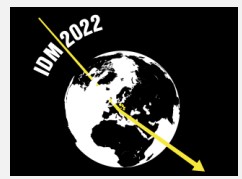

## Abstract

The CRESST experiment observes an unexplained excess of events at low energies. In the current CRESST-III data-taking campaign we are operating detector modules with different designs to narrow down the possible explanations. In this work, we show first observations of the ongoing measurement, focusing on the comparison of time, energy and temperature dependence of the excess in several detectors. These exclude dark matter, radioactive backgrounds and intrinsic sources related to the crystal bulk as a major contribution.

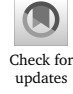

**1** Max-Planck-Institut für Physik, D-80805 München, Germany
**2** Institut für Hochenergiephysik der Österreichischen Akademie der Wissenschaften,
A-1050 Wien, Austria
**3** Atominstitut, Technische Universität Wien, A-1020 Wien, Austria
**4** INFN, Laboratori Nazionali del Gran Sasso, I-67100 Assergi, Italy
**5** LIBPhys-UC, Departamento de Fisica, Universidade de Coimbra,
P3004 516 Coimbra, Portugal
**6** Comenius University, Faculty of Mathematics, Physics and Informatics,
84248 Bratislava, Slovakia
**7** Physik-Department, Technische Universität München, D-85747 Garching, Germany
**8** Walther-Meißner-Institut für Tieftemperaturforschung, D-85748 Garching, Germany
**9** Eberhard-Karls-Universität Tübingen, D-72076 Tübingen, Germany



**10** Department of Physics, University of Oxford, Oxford OX1 3RH, United Kingdom
**11** GSSI-Gran Sasso Science Institute, I-67100 L'Aquila, Italy
**12** Dipartimento di Ingegneria Civile e Meccanica, Università degli Studi di Cassino
e del Lazio Meridionale, I-03043 Cassino, Italy

## Contents

## 1 Introduction

CRESST (Cryogenic Rare Event Search with Superconducting Thermometers) is a dark matter (DM) direct detection experiment located at the Laboratori Nazionali del Gran Sasso (LNGS) in Italy. It uses single crystals as cryogenic calorimeters which are equipped with W transition-edge-sensors (TES) and operated at temperatures of $\mathcal{O}(15\,\mathrm{mK})$. In our first CRESST-III data-taking campaign we achieved a detection threshold of 30.1 eV which allowed probing DM particles with a mass as low as 160 MeV/c$^2$. In the previously inaccessible recoil energy range below 200 eV we measured an unexpected signal. Due to varying shapes and rates in different detector modules, DM as a single origin of this excess signal was excluded [1,2].

Steep rises of the event rate in the sub-keV energy regime exceeding the expected backgrounds are also observed by other low-threshold cryogenic experiments exploiting either phonon signal collection (EDELWEISS [3, 4], NUCLEUS [5, 6], SuperCDMS-HVeV [7, 8], SuperCDMS-CPD [9]) or charge readout via CCD-sensors (DAMIC [10] and SENSEI [11]). A recent summary and comparison of their observed spectra, collected in the scope of the first EXCESS Workshop, is reported in [12]. Since the excess events represent a dominating feature in the region of interest of low-threshold DM and coherent elastic neutrino-nucleus scattering (CE$\nu$NS) experiments, they are the main limitation for their further sensitivity improvement. Therefore, identifying their origin and reducing their impact is currently of utmost importance for the field.

In order to test various hypotheses for the origin of the low energy excess (LEE) in CRESST, we applied several modifications to the detector module design in the current data-taking campaign. We present the energy spectra as well as the temperature and time dependence of

the LEE rate measured by different detector modules. From these observations we can exclude several origins as a source of the LEE.

## 2 Detector design and modifications

The standard design of a CRESST-III module is shown in Fig. 1. It consists of a bulk target crystal (usually scintillating $CaWO_4$) with a size of (20x20x10) $mm^3$ which is read out by a W-TES and held by three sticks (also $CaWO_4$). The main absorber is accompanied by a (20x20x0.4) $mm^3$ Silicon on Sapphire (SOS) wafer, equipped with a W-TES, which usually acts as a light detector. To enhance the light collection, the surfaces facing the detectors are covered by a reflective and scintillating Vikuiti™ foil.

We applied several modifications to the standard module design in this measurement campaign. To study effects related to the target material, we used different crystals for both the bulk and wafer detectors. The bulk detectors are made from Si, $Al_2O_3$ (Sapphire), $LiAlO_2$ and $CaWO_4$. The wafer detectors are made from SOS or Si. One of the $CaWO_4$ crystals, TUM93A, was grown at Technical University of Munich from high-purity powder and with reduced intrinsic stress [13]. To study the effect of the detector holding on the LEE, we mostly replaced the $CaWO_4$ sticks by Cu sticks. The module *Comm2* ($CaWO_4$ absorber) is held by bronze clamps with a broader contact area to exclude effects originating from the point-like connection to the holding sticks. Scintillation light coming from the holding sticks or the foil surrounding the detectors could cause LEE events in the target crystal that are not seen by the light detector. In addition to replacing the scintillating holding sticks, we also removed the scintillating foil in several detectors. While $CaWO_4$, $LiAlO_2$ and $Al_2O_3$ are scintillating crystals, we constructed a fully non-scintillating module from only non-scintillating Si for both bulk and wafer detectors in the *Si2* module. Consequently, the wafer detector was not tuned for detecting scintillation light, but coincident events with the main absorber.

To allow for an accurate and fast energy calibration, we introduced low-activity $^{55}$Fe sources to all modules, which are coated by a layer of glue and gold to reduce the emission of Auger electrons and to make them light-tight.

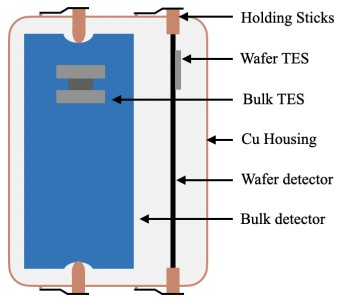

Figure 1: Schematic drawing of the main components of a CRESST-III module (not to scale). The bulk and wafer detector are enclosed by a copper housing and held by sticks. Both detectors are equipped with a W-TES.

Tab. 1 shows an overview of the different configurations for all modules studied in this paper. In this work we present results for the bulk detectors in the modules *TUM93A*, *Comm2*, *Sapp1,2* and *Li1*. For the *Si2* module we show the results for the wafer detector, as this allows a comparison of the energy spectra measured by detectors with different geometries and masses.

## 3 Data analysis

The analysis of the different modules follows the same approach as first described in [1]. In this work we give an overview of the most important steps of the analysis. A detailed description

Table 1: Overview of different module configurations shown in this work. For the *Si2* module, the wafer detector was analysed, for all other detectors, the bulk detector was investigated.

| Module | Target | Holding | Foil | Mass (g) | Threshold (eV) |
|--------|--------|---------|------|----------|----------------|
| Si2 | Si | Cu | No | 0.35 | 10 |
| Sapp1 | $Al_2O_3$ | Cu | No | 16 | 157 |
| Sapp2 | $Al_2O_3$ | Cu | No | 16 | 52 |
| Li1 | $LiAlO_2$ | Cu | Yes | 11 | 84 |
| TUM93A | $CaWO_4$ | 2 Cu + 1 $CaWO_4$ | Yes | 24 | 54 |
| Comm2 | $CaWO_4$ | Bronze Clamps | No | 24 | 29 |

of the individual analysis steps is published in [14] for the *Li1* module. For an illustration of the analysis procedure the detector *TUM93A* is used as an example in the following.

We record the output of each detector with a continuous DAQ to obtain a dead-time free stream of data, which can be further processed offline. In a first step we use an optimum filter [15] to maximize the signal-to-noise ratio. This filter is constructed using an averaged pulse from a sample of recorded particle interactions in the absorber (particle template) and a noise power spectrum to create a weighting function in frequency space. The filter is designed to preserve the pulse heights of the signals, so the filter amplitude can directly be used as an estimator for the signal amplitude. The threshold at which we trigger the filtered data is defined by the choice of accepting one noise trigger event per one kg day of exposure, which can be determined using a large set of empty noise traces, following [16].

We then remove time intervals of unstable operating conditions as well as coincidences with the CRESST muon veto and non-standard pulse shapes, e.g. pileup or electronic artifacts, from the triggered data. For modules with scintillating crystals we select only events originating from recoils in the main absorber (bulk), excluding events that triggered only in the light detector (wafer).

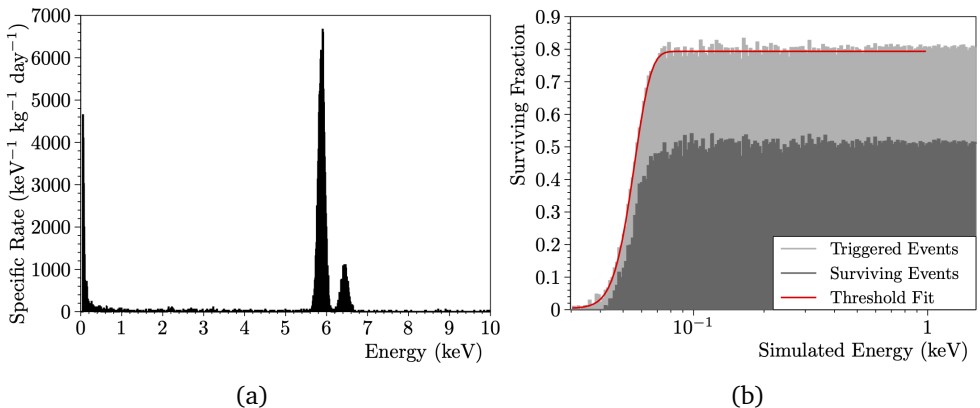

(a)  (b)

Figure 2: (a) Cleaned energy spectrum of detector *TUM93A*, showing the calibration lines of the $^{55}$Fe source and the excess at low energies. (b) In light grey: Fraction of events surviving the trigger condition, fit by an error function to verify the threshold in units of energy. In dark grey: Fraction of events surviving the selection criteria.

For the energy calibration of the selected data we use the $K_\alpha$ and $K_\beta$ lines of the $^{55}$Fe decay at 5.89 keV and 6.49 keV [17], respectively. Fig. 2a shows the cleaned and calibrated energy spectrum of *TUM93A*.

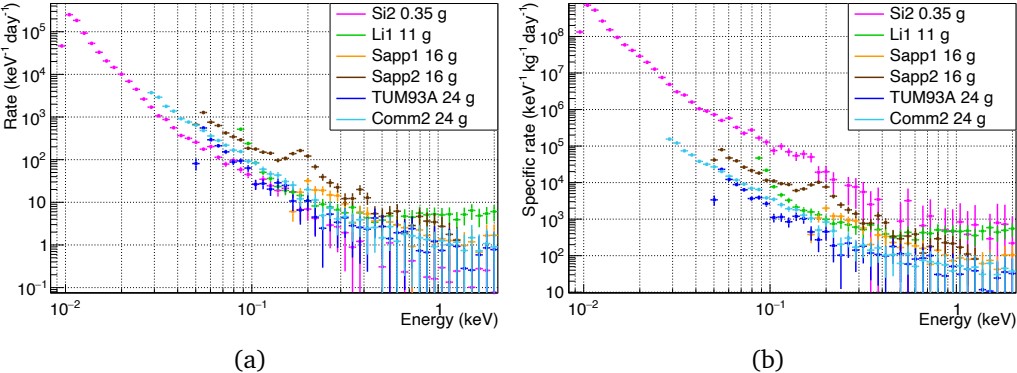

Figure 3: Energy spectra for all detector modules analysed in this work. All spectra are corrected by their respective survival probabilities. (a) shows the spectra scaled by measuring time, and (b) additionally scaled by mass.

Our event selection serves to remove all artifacts and pulses for which we cannot reconstruct the amplitude correctly. We adjust the selection criteria individually for each detector. For a comparison of the results, the energy dependent trigger and selection survival probabilities of real events in the different analyses have to be determined. For this we superimpose artificial signals to the entire stream of recorded data, randomly distributed over time with energies uniformly distributed over the analysed energy range. After processing them with the same analysis pipeline as used for real particle events, we can calculate the fraction of surviving events at each simulated energy. This is used as an estimation of the probability of valid events to survive the trigger (light grey) and selection criteria (dark grey) shown in Fig. 2b. The maximum trigger probability of ∼80 % is due to the dead time introduced by heater pulses injected into the detectors for stabilisation and calibration purposes. Fitting an error function to the trigger probability provides a verification of the trigger threshold in units of energy. The dark grey histogram shows the survival probability of valid events after all selection criteria were applied to the data.

For all modules shown in this work, two independent analyses are performed. For data processing and analysis we use a collaboration internal package "CAT" and the publicly available Python package "Cait" [18].

## 4 Observations

In this section we summarise the observations concerning the LEE seen in all CRESST-III detector modules listed in Tab. 1. A typical CRESST measurement campaign consists of the detector setup period including a calibration with a $^{57}$Co source, followed by a background data-taking period and a neutron calibration. In the current data-taking campaign, a second set of background data was additionally taken to further investigate the time evolution of the LEE.

### 4.1 Energy spectra

In this section we focus on the background data taken in the period from 11/2020 to 08/2021 from here on named BCK data set. Since two detectors had a change in their operating conditions in 02/2021, we choose a reduced data set starting in 02/2021 of 105.4 days to compare the energy spectra measured by various modules in the same time period. Fig. 3 shows time-averaged rate spectra corrected by the respective survival probabilities for all modules from

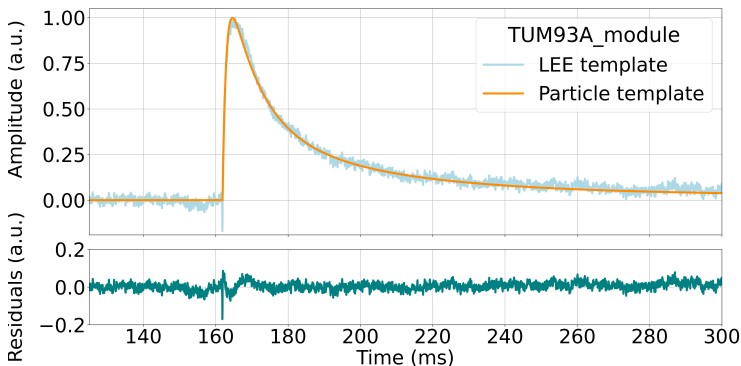

Figure 4: Top: A pulse template for particle interactions in the main absorber (orange) and a template built from LEE events (light blue) for the *TUM93A* detector. The LEE template is obtained by averaging events with energies that do not exceed the threshold by more than two times the energy resolution (54-70 eV for *TUM93A*). Bottom: Difference between both templates.

Tab. 1. Fig. 3a shows the spectra scaled only by the measuring time, while in Fig. 3b the spectra are additionally scaled by the absorber mass. The error bars represent the statistical uncertainties.

We observe a sharp rise of events below a few hundred eV in all detectors, independent of the target materials and holding structures. Furthermore, deviations from a counting rate that decreases uniformly with increasing energy occur at about 180 eV in *Sapp2*, and similarly, albeit with less significance and at lower energies, in *Si2* and *TUM93A*. The origin of this bump-like feature is under investigation. The LEE between different modules neither agrees in rate, which differ up to one order of magnitude in the 60-120 eV range, nor in the specific rate, which is the rate scaled by the mass of the detectors, differing by up to two orders of magnitude. As an example, the *Si2* detector has the lowest rate, but the highest specific rate (by nearly one order of magnitude). Hence, a common single particle origin (e.g. DM or external radiation) of these events is disfavoured.

To exclude that the LEE events are caused by noise fluctuations or artifacts, we build a LEE pulse template by averaging all pulses from the lowest energy region (with energies that do not exceed the threshold by more than two times the energy resolution) of each detector. The top part of Fig. 4 shows a comparison of the LEE template with a noise-free particle template used for the analysis (see section 3) for the *TUM93A* detector module. Low values of the residuals of the comparison confirm that the LEE consists of valid pulses with the same pulse shape as particle recoil events. We observe similar levels of residuals for the other CRESST detectors.

## 4.2 Time dependence of the LEE rate

In the BCK data-taking period, we observe a decay of the LEE rate in all detector modules. For the comparison of the LEE rate in different detectors, we select a common energy range from 60 eV to 120 eV (excluding *Sapp1* and *Li1* modules which have higher thresholds). The time dependence of the LEE rate for the BCK data set is shown in the top of Fig. 5 for a time period from 90 to 380 days since the first cool-down of the cryostat. Each data point shows the measured count rate within one week (livetime roughly 150 h), corrected with their corresponding survival probabilities and livetime. The error bars indicate the statistical uncertainties.

A neutron calibration was performed at the end of the BCK data set. To verify that the LEE rate was not affected by the neutron calibration, another short background data set was taken afterwards. We observe no impact on the rate, as can be seen in Fig. 5. Following

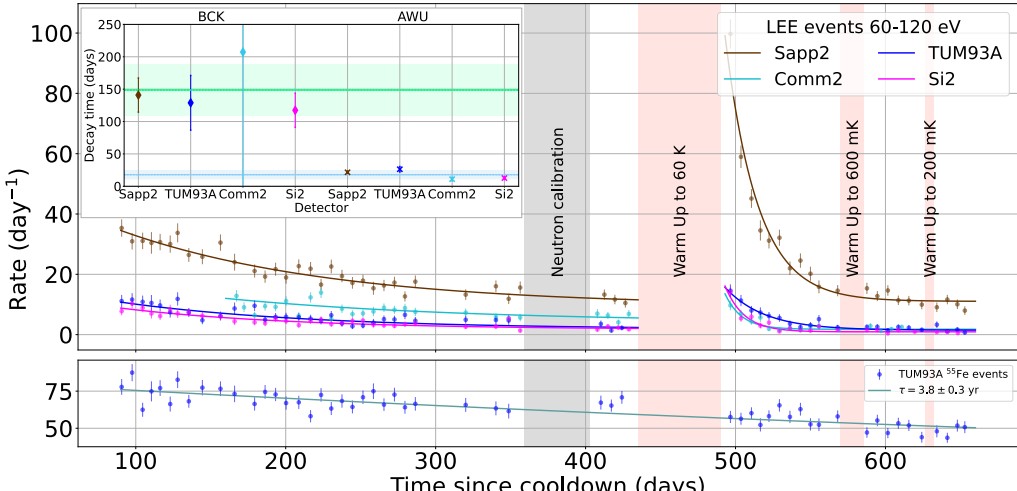

Figure 5: Top: Time evolution of LEE rates (60-120 eV) in different detector modules for BCK (90-380 days) and AWU (495-670 days) data sets. Solid lines show the fitted functions ($R(t) = A \cdot exp(-\frac{t}{\tau}) + C$). The inset plot shows the resulting decay times from fits for the BCK (diamond) and AWU (cross) periods. Dashed lines and shaded areas represent the mean values and standard deviations of decay times across four modules within the BCK (green) and AWU (blue) periods. Bottom: Decay of the $^{55}$Fe event rate measured in the *TUM93A* module with a decay time agreeing with the literature value of 3.9 yr [17].

discussions at the previous EXCESS Workshop, we interrupted the data-taking for a dedicated investigation of time and temperature dependence of the LEE. During this break the cryostat was warmed up to ∼60 K and then cooled down again to the base temperature of $\mathcal{O}(15\,\text{mK})$. The data-taking was resumed in 12/2021. This data set is called the After-Warm-Up (AWU) data set in the following.

We observe a much higher rate in the AWU data set, directly after the detectors went back to their operation point (see Fig. 5 beyond t=490 days), decaying faster than in the BCK data set. We also performed two additional warm-up tests with smaller temperature increase during the AWU data set, once to 600 mK and afterwards to 200 mK. These later warm-up tests do not influence the LEE rate (Fig. 5). To quantify this observation we fit the data points with an exponentially decaying rate $R(t) = A \cdot exp(-\frac{t}{\tau}) + C$ for each detector. The decay times $\tau$ of each fit are shown in the inset plot of Fig. 5. The uncertainties of the decay time of the *Comm2* module are higher than in the other modules due to a reduced time range, caused by a later start of the data-taking. The average decay time for the AWU period across different modules is $(18 \pm 7)$ days while for the BCK period it is as high as $(149 \pm 40)$ days. In the bottom of Fig. 5, we show the decay time of the $^{55}$Fe events as a reference in one of the modules with a much higher value of $(3.8 \pm 0.3)$ yr, which agrees with the literature value of 3.9 yr (corresponding to a half-life of 2.7 yr [17]).

A similar observation but in the energy range above 5 keV was made by the EDELWEISS collaboration for the heat-only events [19]. The reset of the LEE rate by a warm-up of the detectors excludes external as well as intrinsic radioactive origins as a major contribution. Also, it is another confirmation for the exclusion of DM as a single origin of the excess.

## 5 Summary

In this work we present several new observations on the LEE based on the current CRESST-III data-taking campaign. Thanks to our TES-based technology it is possible to study the origin of the excess events at energies below 100 eV for many different target materials. None of the modifications that were applied to the standard CRESST-III detector modules have a significant impact on the presence of the LEE.

The excess is present in all investigated detector materials and does not scale by mass. In addition, the LEE events have the same pulse shape as the particle recoil events. We observe an exponential decay in the LEE rate in each module with a decay time not compatible with that of the $^{55}$Fe source installed in the modules. The rate increases after a warm-up of the modules to 60 K and decays faster afterwards. Two additional warm-up tests to lower temperatures of 600 mK and 200 mK do not show an effect on the LEE.

These observations lead to the exclusion of several hypotheses on major contributions. DM interactions and external radioactivity can be excluded due to a varying absolute rate in different modules and the effect of the warm-up, which in addition excludes intrinsic radioactive backgrounds, since this would not re-scale during a warm-up of the detectors. Scintillation light as a source for the LEE can be excluded as the latter is also present in the non-scintillating modules.

Despite the LEE being present in all modules, regardless of the crystal material or the different holdings, intrinsic crystal effects (e.g. stress release), sensor- (e.g. from the TES film deposition) and holding-related origins are still possible options and are subject of further investigations within the CRESST collaboration. The hypotheses considered in this work are by no means a complete list of possible origins of the LEE, additional ideas can be found in [12].

## Acknowledgements

We are grateful to LNGS for their generous support of CRESST. This work has been funded by the Deutsche Forschungsgemeinschaft (DFG, German Research Foundation) under Germany's Excellence Strategy – EXC 2094 – 390783311 and through the Sonderforschungsbereich (Collaborative Research Center) SFB1258 'Neutrinos and Dark Matter in Astro- and Particle Physics', by the BMBF 05A20WO1 and 05A20VTA and by the Austrian science fund (FWF): I5420-N, W1252-N27. FW was supported through the Austrian research promotion agency (FFG), project ML4CPD. SG was supported through the FWF project STRONG-DM (FG1). The Bratislava group acknowledges a partial support provided by the Slovak Research and Development Agency (project APVV-15-0576). The computational results presented were partially obtained using the Vienna CLIP cluster and the MPCDF Munich. We found discussions with the colleagues from other collaborations concerning the LEE, during the EXCESS Workshops and beyond, very helpful.

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
