# Peer review of "Latest observations on the low energy excess in CRESST-III"

_SciPost Physics Proceedings, doi:SciPost Phys. Proc. 12, 013 (2023)_

## Round 1 · Referee Report · Anonymous (Referee 1) · 2022-10-6

Report

These proceedings describe highly interesting and important measurement campaigns conducted by the CRESST collaboration with the dedicated goal to shed some light on the composition of the "low energy excess" observed by multiple solid state low threshold direct DM experiments . These excesses are a current limitation towards the lightest DM masses these devices could technically probe. The results that are presented are thus not only relevant for CRESST but to the entire low threshold community. The authors studied the rate behavior as a function of energy and as a function of time using a set of mostly different, modified CRESST detector modules. Their findings allow the exclusion of various hypotheses about the main contribution to the excess observed in CRESST, which is a major step forward. In particular it can be excluded that DM interactions or external and intrinsic radioactivity are the main source.

The proceedings are very well-written, supported by convincing plots. The content is highly relevant for the community. I thus recommend publication and I wish the collaboration all the best for their further investigations.

I have only one short question and two minor fixes. None of which requires my re-evaluation.

Requested changes

Sec. 3 * Just a question. The authors say "The threshold at which we trigger the filtered data is defined by the choice of accepting one noise trigger event per one kg day of exposure, following". I assume this was evaluated using random noise traces? And if so, how was the accidental appearance of true low energy physics pulses dealt with? An extra sentence for clarification would be nice but I wouldn't insist on it because this threshold really is a mere choice. In the end what matters is the actual turn-on curve as determined and presented by the authors. * "The dark grey curve shows the survival probability of valid events after all selection criteria were applied to the data." This sentence refers to Fig. 2b but that figure seems to be missing that curve. It only shows the respective histogram to which one could fit a function resulting in the mentioned PDF. So I'd suggest the authors to either add the curve to the figure or remove the sentence from the text.

Sec. 4 * Typo close to the end of the section: "In the bottom of figure Fig. 5, " -> "In the bottom of Fig. 5"

  • validity: top
  • significance: top
  • originality: top
  • clarity: top
  • formatting: perfect
  • grammar: perfect

Author:  Dominik Fuchs  on 2022-10-17  [id 2931]

(in reply to Report 1 on 2022-10-06)
Category:
answer to question

Thank you a lot for your comments. We will implement these minor points and resubmit in the next days. Just to answer your question about the threshold determination: Yes, we use a large set of randomly drawn empty noise traces. We also apply some selection criteria that remove any pulses from this data set.

---

## Round 1 · Referee Report · Anonymous (Referee 2) · 2022-10-7

Strengths

The is a well written manuscript that presents and explains the data in a clear and straightforward manner. The conclusions are valid.

Report

The is manuscript's is of high quality and I recommend that it be accepted for publication.

---

## Round 2 · Referee Report · Anonymous (Referee 1) · 2022-11-2

Report

My comments were very minor to begin with and all of them have been addressed. The proceedings are well-written, clear, and highly relevant and I recommend their acceptance.

---

## Round 2 · Referee Report · Anonymous (Referee 2) · 2022-11-16

Strengths

The paper reports on result s relevant to a currently active area in dark matter searches. The results are presented in a clear and and coherent manner.

Weaknesses

The paper contains discernable weaknesses that need to be addresses.

Report

I recommend that this manuscript be accepted for publication.

---

## Editorial Decision

published